# Artificial Intelligence in Lung Cancer Screening: The Future Is Now

**DOI:** 10.3390/cancers15174344

**Published:** 2023-08-30

**Authors:** Michaela Cellina, Laura Maria Cacioppa, Maurizio Cè, Vittoria Chiarpenello, Marco Costa, Zakaria Vincenzo, Daniele Pais, Maria Vittoria Bausano, Nicolò Rossini, Alessandra Bruno, Chiara Floridi

**Affiliations:** 1Radiology Department, Fatebenefratelli Hospital, ASST Fatebenefratelli Sacco, 20121 Milano, Italy; michaela.cellina@asst-fbf-sacco.it; 2Department of Clinical, Special and Dental Sciences, University Politecnica delle Marche, 60126 Ancona, Italy; laura.cacioppa@gmail.com (L.M.C.); nicolorossini44@gmail.com (N.R.); alessandrabruno92@gmail.com (A.B.); 3Division of Interventional Radiology, Department of Radiological Sciences, University Hospital “Azienda Ospedaliera Universitaria delle Marche”, 60126 Ancona, Italy; 4Postgraduation School in Radiodiagnostics, Università degli Studi di Milano, 20122 Milan, Italy; maurizioce.md1@gmail.com (M.C.); vittoria.chiarpenello@unimi.it (V.C.); marco.costa@unimi.it (M.C.); zakaria.vincenzo@unimi.it (Z.V.); daniele.pais@unimi.it (D.P.); mariavittoria.bausano@unimi.it (M.V.B.); 5Division of Radiology, Department of Radiological Sciences, University Hospital “Azienda Ospedaliera Universitaria delle Marche”, 60126 Ancona, Italy

**Keywords:** artificial intelligence, lung cancer, lung cancer screening, computer-aided detection, lung nodule, virtual biopsy, radiomic, deep learning, personalized medicine, personalized screening programs, segmentation

## Abstract

**Simple Summary:**

Lung cancer is a widespread malignant tumour with a high mortality and morbidity rate and is frequently diagnosed in the middle and late stages when few therapies are available. Lung cancer screening allows an early-stage diagnosis and more effective therapies. Artificial intelligence (AI) plays a key role in lung cancer screening workflow for early diagnosis. Particularly, in low-dose computed tomography for screening programs, AI further reduces radiation dose maintaining an optimal image quality. AI also allows risk stratification and subsequent screening personalization. A computer-aided detection (CAD) system helps in lung nodule detection with a high sensibility, reducing imaging time interpretation. AI is additionally applied in nodule characterization (benign or malignant), using different approaches. This narrative review aims to provide an overall view of all possible AI applications in lung cancer screening.

**Abstract:**

Lung cancer has one of the worst morbidity and fatality rates of any malignant tumour. Most lung cancers are discovered in the middle and late stages of the disease, when treatment choices are limited, and patients’ survival rate is low. The aim of lung cancer screening is the identification of lung malignancies in the early stage of the disease, when more options for effective treatments are available, to improve the patients’ outcomes. The desire to improve the efficacy and efficiency of clinical care continues to drive multiple innovations into practice for better patient management, and in this context, artificial intelligence (AI) plays a key role. AI may have a role in each process of the lung cancer screening workflow. First, in the acquisition of low-dose computed tomography for screening programs, AI-based reconstruction allows a further dose reduction, while still maintaining an optimal image quality. AI can help the personalization of screening programs through risk stratification based on the collection and analysis of a huge amount of imaging and clinical data. A computer-aided detection (CAD) system provides automatic detection of potential lung nodules with high sensitivity, working as a concurrent or second reader and reducing the time needed for image interpretation. Once a nodule has been detected, it should be characterized as benign or malignant. Two AI-based approaches are available to perform this task: the first one is represented by automatic segmentation with a consequent assessment of the lesion size, volume, and densitometric features; the second consists of segmentation first, followed by radiomic features extraction to characterize the whole abnormalities providing the so-called “virtual biopsy”. This narrative review aims to provide an overview of all possible AI applications in lung cancer screening.

## 1. Introduction

Artificial intelligence (AI) is a term coined by John McCarthy in 1956 with the aim of describing the approach of using computers and technology to simulate intelligent behaviour and critical thinking comparable to a human [1]. Several improvements in different medical fields can stem from its potential in diagnosis, management, and treatment outcome prediction through the analysis and interpretation of complex data [2].

Radiology represents the medical field in which AI techniques have been earlier incorporated [3]. Lung cancer (LC) is the main cause of worldwide malignancy-related deaths, diagnosed in 1.8 million people annually. There are 1.6 million people who die each year due to the disease [4] and the net five-year or more survival rate is 13.8%.

In Italy, LC is the primary cause of cancer death (20% of all cancer deaths in 2018) and the third most prevalent neoplasm (11% of all cancers in 2018) [5,6].

Studies on lung cancer epidemiology and related risk factors influenced tertiary lung cancer prevention, as it is estimated that 85–90% of LC is associated with cigarette smoking in Italy [7]. Smoking cessation decreases LC risk after 10 years after quitting [6,8] and is the intervention for primary prevention of LC [6,9]. A high percentage of isolated lung nodules may be potentially curable early lung cancers, although they are mostly benign [10]. Furthermore, 75% of lung cancer is detected only in its advanced stages [11] and, moreover, it has very heterogeneous imaging and histopathology that makes the choice of the best treatment option difficult.

In this narrative review, we aim to provide an overview of the possible application of AI in lung cancer screening.

We performed a literature search using the following key words: “lung cancer screening”, “personalized lung cancer screening”, “personalized lung cancer screening program”, “lung nodule detection”, “lung nodule characterization “, “low-dose chest CT reconstruction” combined with “artificial intelligence”, “deep learning”, or “machine learning” on the PubMed and Google Scholar.

## 2. The Screening Rationale

Studies such as the National Lung Screening Trial (NLST) have demonstrated how screening for lung cancer a vital tool in high-risk individuals is because the early stages of the disease are completely asymptomatic or cause minimal non-specific symptoms [11].

In 2010, an evaluation of NLST data showed a statistically significant 20% reduction in lung cancer mortality in a high-risk adults group who received three consecutive annual lung cancer screening exams with low-dose computed tomography (LDCT) versus another high-risk adults group who received three consecutive annual chest X-rays [12].

Results from the NLST study were published in 2011, providing the first evidence from a prospective randomized controlled trial that lung cancer screening reduces lung cancer deaths in a high-risk population. [13].

Also according to other several randomized controlled trials, screening with LDCT is the main intervention for secondary prevention of LC and decreases LC mortality by 20–30%, especially in women [13,14,15,16,17].

Lung cancer is still the first cause of cancer-related death in developing countries and the high mortality and morbidity of lung cancer diseases need improvement and modification of lung cancer screening techniques. All individuals cannot be screened for lung cancer, but at least higher-risk people should be [18]. CT screening could improve cure rates by routinely detecting millimetric potentially malignant lung nodules in high-risk populations [19,20].

Early detection, diagnosis, and better management of lung cancer can be favoured by advanced technologies to improve the outcome, as a late diagnosis is linked to a poor prognosis.

Radiologists had a high workload using CT techniques to evaluate, characterize, and detect lung nodules [21]. Technological advances could help radiologists reduce the amount of work and increase the sensitivity of screening keeping false positive rates low. Moreover, thanks to an improvement in technologies in lung cancer screening, morbidity and mortality associated with lung malignancies could be reduced by increasing early diagnosis and appropriate management rates [22].

Various Italian trials about lung screening are present in the literature and a common document from the Italian College of Thoracic Radiology has been available since 2022 [6]. Promising results about lung screening are revealed in three Italian pivotal studies that are the Multicentric Italian Lung Detection study (MILD), the ITALUNG Trial, and the detection and screening of early lung cancer with novel imaging technology (DANTE) study. Moreover, non-randomized trials (COSMOS and BioMILD) show the importance of blood biomarkers integration with LDCT every 3 years [6,23,24].

Currently, the United States and China are the only countries that offer lung cancer screening everywhere, whereas countries like the Netherlands and the UK offer screening only in some locations. In Europe, a shared program of lung cancer screening has not started yet despite the availability of several shared guidelines [25,26]. The incorporation of LCS via LDCT into the public Italian healthcare system is currently being discussed (for example, Livelli Essenziali di Assistenza, LEA) [27].

The most recent version of the American Cancer Society (ACS) lung cancer screening guideline (2018) is being taken down and has to be updated yearly in relation to new scientific evidence about screening [28]. These guidelines support yearly lung cancer screening using LDCT scans for adults aged 50 to 80 who are in excellent health, presently smoke or have quit within the last 15 years, and have a 20-pack-year smoking history. Furthermore, it is critical that people who will be screened receive adequate counselling to quit smoking if they currently smoke, that their doctors are aware of the potential benefits, limitations, and harms of screening with LDCT scans, and that they are directed to lung cancer screening and treatment-experienced centres. [6,28]. Currently, the European Commission recommends introducing lung cancer screening for current and ex-smokers who have quit smoking within the previous 15 years, are aged 50 to 75 years, and have a smoking history of 30-pack-years [25,26,27,28].

## 3. AI Terminology

Machine learning is a scientific field dedicated to the study of how computers acquire knowledge from data. It emerges at the convergence of statistics, which aims to uncover relationships within data, and computer science, which focuses on developing efficient computational algorithms. This amalgamation of mathematics and computer science is motivated by the distinctive computational hurdles encountered when constructing statistical models from extensive datasets, which may consist of billions or trillions of data points [29,30,31,32].

Computer learning methods are conveniently classified into distinct categories, including supervised learning and unsupervised learning.

The main difference between supervised machine learning and unsupervised machine learning is in the type of data used for training the models and the learning objectives.

Supervised machine learning relies on the use of labelled training data, which means input data that are associated with correct labels or responses [30,31,32]. The goal is to train the model to make predictions or decisions similar to new data. For example, in training a cat image classification model, each image is provided with the corresponding label “cat” or “non-cat”. The model learns from the labelled data and tries to generalize to make accurate predictions on new unseen data [31,32].

On the other hand, unsupervised machine learning is based on the use of unlabeled training data, which means input data that do not have associated labels or correct responses [32]. The main objective is to discover patterns, structures, or clusters hidden within the data. In this case, the model does not receive explicit information on what to look for but must learn autonomously to recognize meaningful patterns or create groupings of similar data [32,33]. For example, a clustering model could automatically group similar data without having information about the labels of the groups. In summary, supervised machine learning requires labelled data and focuses on predicting or classifying new data based on the information learned from labelled training data. Unsupervised machine learning uses unlabeled data to discover hidden patterns or clusters in the data without specific prediction goals [32,33].

The primary distinction between supervised machine learning and unsupervised machine learning holds significant implications within the medical field, particularly in the context of data analysis and pattern recognition [34].

Supervised machine learning in the medical domain relies on the utilization of meticulously labelled training data, encompassing input data that is associated with accurate labels or known responses. The overarching objective involves training the model to effectively predict or classify new medical data, drawing insights from the learned patterns derived from the labelled training data. For instance, when training a model to classify medical images, each image would be accompanied by the corresponding label denoting the presence or absence of a specific condition. The model subsequently assimilates knowledge from the labelled data, enabling it to generalize and make accurate predictions on unseen medical data [32,34,35].

Contrarily, unsupervised machine learning methods in the medical context involve utilizing unlabeled training data, where input data do not possess predefined labels or known responses. The primary goal revolves around the discovery of hidden patterns, structures, or clusters within medical data, without specific prediction objectives. In this scenario, the model autonomously learns to identify meaningful relationships or create groupings of similar medical data without reliance on explicit labels. For example, unsupervised clustering techniques can be employed to uncover natural groupings within patient data, aiding in the identification of potential disease subtypes or patient cohorts [32,34,36]. Understanding the distinction between supervised and unsupervised machine learning is paramount within the medical field. Supervised methods facilitate the development of predictive models that can assist in diagnosing medical conditions, triaging patients, or predicting treatment outcomes. In contrast, unsupervised techniques offer valuable insights into the inherent structures within medical data, uncovering hidden relationships that may have clinical significance and aiding in exploratory data analysis.

The choice between supervised and unsupervised machine learning approaches depends on the availability and nature of labelled training data, as well as the specific objectives of the medical analysis. Both methodologies play pivotal roles in advancing medical research, diagnosis, and treatment, offering complementary perspectives for comprehensive data exploration within the healthcare domain [37]. Machine learning has revolutionized radiology by revolutionizing the accuracy and efficiency of various tasks. Radiologists, who specialize in interpreting and diagnosing medical images like X-rays, CT scans, MRI scans, and ultrasound images, have greatly benefited from the integration of machine learning algorithms into their practices.

One of the notable applications of machine learning in radiology is image classification and segmentation [38]. By utilizing techniques such as convolutional neural networks, radiologists can automatically classify anatomical structures and identify abnormalities in medical images. This technology has proven invaluable in detecting tumours, identifying lung nodules and accurately segmenting organs.

Furthermore, machine learning algorithms contribute to quality control and image enhancement: by reducing noise, enhancing details, and standardizing image acquisition protocols, these algorithms improve the quality of medical images. This ensures consistent interpretation and enhances the overall diagnostic process.

It is crucial to emphasize that while machine learning algorithms have shown tremendous potential in radiology, they are designed to complement radiologists rather than replace them. Radiologists’ expertise and clinical judgment remain indispensable for accurate diagnosis and effective patient care [31].

Deep learning, a type of machine learning, has the goal of training artificial neural networks with many layers to recognize hierarchical data representations [39]. It is inspired by the structure and function of the human brain, specifically the interconnected network of neurons [34,38,39].

In deep learning, artificial neural networks, also known as deep neural networks, are composed of multiple layers of interconnected nodes or neurons. Each neuron receives input signals, applies a mathematical operation to them, and produces an output signal. The layers in a deep neural network are typically organized into an input layer, one or more hidden layers, and an output layer.

Deep learning algorithms learn by iteratively adjusting the weights and biases of the neurons in the network based on the provided training data [40,41]. This process is known as training or learning. During training, the network learns to recognize patterns and extract relevant features from the input data. The deeper layers in the network learn increasingly complex representations of the data, which enables the network to capture intricate relationships and make accurate predictions.

One of the key advantages of deep learning is its ability to automatically learn features from raw data, eliminating the need for manual feature engineering.

Deep learning has achieved remarkable success in various applications. For example, in computer vision, deep learning models have achieved state-of-the-art performance in tasks like object detection, image classification, and image segmentation [40,41]. This section is divided into subheadings to give a clear and succinct explanation of the experimental findings, their interpretation, and any possible experimental inferences.

## 4. AI Applications in Lung Cancer Screening

### 4.1. Personalized Screening Programs

Numerous organizations support the use of individualized risk-stratified techniques for lung cancer screening [25,41,42,43]. There is ongoing discussion over the best approaches and requirements for integrating risk stratification based on quantitative estimates of risk in lung cancer screening programs [25,26]. The literature discussed the effect of the currently proposed risk-based strategies on the benefits and drawbacks to individuals [26,44,45]. Challenges remain in applying risk stratification in practice [46,47,48].

First off, using risk prediction models to determine who is eligible for screening could be a risk stratification strategy in lung cancer screening programs [49,50].

In lung cancer screening, risk prediction is more important than clinical assessment, and different trials demonstrate this, such as the Bach model, [51], the Lung Cancer Risk Assessment Tool (LCRAT) [52], the Lung Cancer Death Risk Assessment Tool (LCDRAT) [52], the Liverpool Lung Project (LLP) model [53,54], and the PLCOm2012 model [55,56].

Screening invitations and risk-communication tactics should also follow a risk-stratified approach [50].

According to studies [57], a large proportion of individuals who are at a high risk of acquiring lung cancer come from socioeconomically disadvantaged families because lung cancer screening is less accessible for them [47,57,58,59].

According to the NLST and NELSON studies [59,60,61], screening findings could be used to determine a person’s individualized screening interval. Given that 90% of CT screening results are negative, an accurate risk stratification is necessary to select the number and the interval of screening CT for every patient [25,62,63]. The cost-efficiency of the program could be increased by lowering the number of CT scans while maintaining program effectiveness [64,65,66]. Additionally, the rising demand for medical imaging has limited radiologists’ ability to interpret lung cancer screening tests in many countries, so fewer CT scans would make adoption easier [67,68,69].

The use of CT screening data to choose appropriate screening intervals is a subject of current research [70]. Based on information from the NLST, Schreuder et al. created a model to prevent 10.4% of all second-round lung screening. This 1-year risk model does not delay lung cancer diagnosis [70].

Higher percentages of examinations may be avoided, but this would mean that a higher percentage of lung cancer diagnoses would have to be delayed. [50]. Robbins et al. integrated people’s pre-screening risk with CT results obtaining a 1-year lung cancer risk model. Moreover, they showed that screening intervals can be delayed but with the risk of lagging some diagnosis. [70,71].

Tammemägi et al. expanded the PLCOm2012 model by incorporating the reclassified Lung-RADS screening results from the NLST study. The study found that positive screening test results were indicative of an increased risk of lung cancer, regardless of the initial PLCOm2012 risk assessment. For individuals whose PLCOm2012 risk exceeded a certain threshold, it was recommended to continue annual screenings. However, for those with PLCOm2012 risk below the threshold, the screening interval could be extended. Interestingly, there were cases where individuals had a high baseline PLCOm2012 risk, and even after three consecutive negative screens, their subsequent lung cancer incidence remained elevated [71,72].

Also, laboratory parameters such as autoantibodies, DNA fragments, microRNA, and other blood-circulating components are studied for their potential role in the risk stratification of patients with lung cancer [73,74].

They could become important in risk prediction models but are still not validated [75].

Ongoing research shows an encouraging and beneficial role of current biomarkers in lung cancer screening. Preliminary results from the Bio-MILD study show that the integration of blood microRNA with CT screening results helps in choosing the correct screening interval [50]. MicroRNA profiles also seem to be important in distinguishing symptomatic lung cancer patients from controls [76] and polygenic risk scores seem to help in identifying a person’s absolute risk [77]. Instead, blood-based biomarkers could help in nodule malignancy characterization [76,77].

It is recommended to explore recent progress and innovations in computer-aided diagnosis and imaging analysis methods to enhance the sensitivity of CT screening and alleviate the workload on radiologists [78].

### 4.2. Image Reconstruction

Low-dose CT scan is the only strategy that has proven to effectively reduce mortality in lung cancer screening in high-risk patients [79].

Despite the development of low-dose CT protocols, Refs. [80,81] repeated scans can expose patients to cumulative radiation risk [82,83,84]. To reduce radiation while maintaining accuracy, there are techniques like model-based iterative reconstruction (MBIR) and hybrid iterative reconstruction (HIR) [84,85]. These reconstruction algorithms are widely used to reduce image noise and artefacts but have limitations in low-dose CT scans.

The deep learning reconstruction (DLR) algorithm represents a new approach to reducing radiation exposure [86]. Pairs of noisy and noise-free images to extract the true information from the noisy ones train the algorithm. DLR reduces noise and improves image quality at the same time, unlike other methods that have trade-offs [86,87,88].

In a study comparing different techniques, DLR performed better than MBIR and HIR in accurately measuring the size of artificial lung nodules in low-dose CT scans. The measurements were performed on known-sized nodules using ultra-low-dose CT. DLR had lower errors in measuring the volume compared to the other methods [87].

DLR also showed better agreement between different observers in making the measurements [88,89]: this means that different doctors or researchers using DLR were more likely to obtain similar results compared to using MBIR or HIR [89,90].

Vendor-agnostic deep learning algorithms can enhance image quality and reduce radiation dose. This is crucial because chest CT scans are in high demand during the COVID-19 pandemic [91]. Another deep learning model (DLM) is ClariCT.AI (ClariPI), which works in post-processing imaging and does not require projection data. Nam et al. demonstrated that for ultra-low-dose CT, vendor-agnostic DLM was better than vendor-specific DLR. However, this study evaluates image quality subjectively at a single radiation dose without considering diagnostic performance [92].

Therefore, DLR is a new reconstruction method that improves the accuracy of measuring lung nodule sizes in low-dose CT scans compared to other techniques like MBIR and HIR. It reduces radiation exposure while maintaining accurate measurements, especially for patients who need long-term follow-up.

### 4.3. CAD System

LDCT has become a common method for lung cancer screening, offering the advantage of early detection of lung nodules. However, the implementation of LDCT poses challenges for radiologists, due to the high burden of examinations to be analysed and reported. The increased noise and reduced spatial resolution associated with low-dose imaging negatively affect the visibility of low-contrast lesions. This can make it difficult for radiologists to accurately identify and interpret small nodules. Distinguishing between benign and malignant nodules is also challenging with low-dose CT images. The potential for error and the possibility of missing nodules, including potential cancerous ones, are concerns associated with manual reading and the sheer volume of cases. It is important to note that LDCT scans typically expose patients to radiation within the range of 1–4 millisieverts [93].

The diagnosis of lung cancer often relies on the identification of lung nodules, which represent important radiological indicators for early detection. However, the treatment of lung cancer nodules can be complex. Radiation therapy is often necessary for lung cancer treatment, with subsequent radiation-induced lung lesions. These injuries are a limiting factor for lung parenchyma study. In this context, computer-aided diagnostic (CAD) systems have a key role for radiologists. CAD systems reduce observational errors and false-negative rates [93] and provide a second opinion in imaging interpretation finalized to diagnosis. By leveraging advanced algorithms and automated analysis, CAD systems help extract additional information from nodules, enhancing the diagnostic process and supporting more accurate decision-making. CAD systems are crucial in assisting radiologists by providing complementary insights and potentially improving the overall quality of care in lung cancer diagnosis and treatment.

Numerous studies have reported how the integration of computer-aided diagnostic (CAD) systems into the diagnostic process can enhance the performance of image diagnosis by reducing inter-observer variability [94]. CAD systems offer several benefits in clinical decision-making. Quantitative support in determining the need for biopsies, significant help in diagnostic examinations, and minimizing unnecessary invasive surgeries such as thoracotomies are just a few of their potentials. Additionally, CAD systems may assist in differentiating between malignant and benign tumours, contributing to more accurate diagnoses. By providing additional analysis and objective data, CAD systems serve as valuable tools in integrating the expertise of radiologists and improving the overall efficiency and reliability of the diagnostic process [93,94,95,96].

#### 4.3.1. Structure of the CAD Systems

CAD systems contain different general components such as data collection and pre-processing. Also, more specific functions finalized for lung analysis are present, such as lung segmentation. Instead, other applications permit precise lung nodule detection, segmentation, and classification, trying to reduce false negatives [95]. Radiological images used by the CAD system are gathered in the data-collecting process. Due to its great sensitivity and relatively low cost, CT is the imaging technique of choice for early nodule detection. The pre-processing stage improves the image quality for the succeeding steps by removing noise, artefacts, and other unnecessary information from the images. Clinicians separate the boundaries of the lung from the surrounding thoracic parenchyma in the CT scans during lung segmentation.

The detection step entails locating the lung mass or nodule, and the false positive reduction step comes next. The false positive phase is a crucial procedure that involves separating the real lung nodules from the candidate nodules that were found. Each lung nodule is separated from the lung parenchyma during the lung nodule segmentation process [96,97]. The nodule characteristics are then measured during the feature extraction stage. The nodule classification stage makes additional use of these characteristics. The nodule is the last and most important part of a CAD system, and it entails distinguishing between benign and malignant nodules.

#### 4.3.2. Data Collection

The CT image acquisition and collection is the crucial step for CAD system development [97]. Public databases are progressively substituting private databases in order to facilitate CAD technology development and allow direct comparison of different systems.

Some public databases are used more frequently than others such as Lung Image Database Consortium and Image Database Resource Initiative (LIDC-IDRI) and Lung Nodule Analysis 2016 (LUNA16). Other two frequently used datasets are the Early Lung Cancer Action Program (ELCAP) and the Automatic Nodule Detection 2009 (ANODE09) [98].

The LIDC-IDRI (Lung Image Database Consortium—Image Database Resource Initiative) is a widely utilized database of lung CT images that aims to facilitate research on computer-aided detection (CAD) technologies for lung cancer. Established by the National Cancer Institute (NCI), the database represents a valuable resource for the detection and diagnosis of lung cancer [96].

The initiative received further support from the Food and Drug Administration (FDA) and the National Institutes of Health (NIH), which led to the creation of the Image Database Resource Initiative (IDRI) in 2004. The IDRI works in conjunction with LIDC to advance its objectives.

Currently, the LIDC-IDRI database includes 1018 chest CT scans with relative lung nodule information given by four expert radiologists that classified lung lesions into three groups in relation to dimensions and characteristics. Moreover, each radiologist reviewed his personal annotations and those of his/her colleagues. These data come from seven academic centres and eight medical imaging companies. In the end, 7371 lesions were considered nodules by at least one radiologist, 2669 of these were nodules ≥ 3 mm, and 928 were nodules for all the radiologists.

The LIDC-IDRI is a large collection of thoracic CT scans and a good database for researchers in order to evaluate CAD algorithms [99].

LUNA16 system refers to a dataset used for training and evaluating deep learning algorithms for medical image analysis.

LUNA16 stands for “LUng Nodule Analysis 2016”, and it is a publicly available dataset created for the purpose of developing computer algorithms for the detection and diagnosis of lung nodules in computed tomography (CT) scans. The dataset was released in conjunction with the LUNA16 Challenge, which aimed to encourage the development of automated algorithms for lung nodule detection.

The LUNA16 dataset is created form the selection of all thin-slice CT scans (less than 3 mm in slice thickness) from the LIDC dataset and contains 888 exams [98].

The scans were anonymized and provided in the form of MHD files. These CT scans contain various types of pulmonary nodules, including malignant and benign nodules of different sizes and shapes [100].

The ELCAP database contains 50 LDCT exams (1.25 mm slice thickness) coming from the International Early Lung Cancer Action Program (I-ELCAP) and the vision and image analysis group at Cornell University [96]. Experienced radiologists have marked the locations of pulmonary nodules within the scans.

The ANODE09 database, established by Nederland Leuvens Longkanker Onderzoek (NELSON), is Europe’s largest CT lung cancer screening database and received a significant contribution from the University Medical Centre at Utrecht. This centre provided 55 CT scans (0.71 mm of average slice thickness) from 50 to 75 aged male heavy smokers. Five scans are annotated and used for training purposes, whereas 50 scans are used only for testing given that are not annotated [96,101].

The Non-Small Cell Lung Cancer (NSCLC) dataset is a wide dataset including 1355 CT exams coming from 211 patients. Of the exams, 144 are axial CT obtained using an automatic segmentation algorithm. The segmentations obtained from the algorithm were further reviewed by thoracic radiologists, each possessing over five years of experience. The annotations for the nodules detected in the images are stored in AIM format, allowing for standardized representation and sharing of the data [96,102].

#### 4.3.3. Accuracy of CAD Systems

Nowadays the utility of AI, through CAD systems, in lung cancer screening for lesion detection and segmentation is widely recognized and several CAD system tools have been validated and are available on the market (Figure 1).

Automatic nodule detection is used to identify lung structures suspected of malignancy.

In particular, deep learning system applications in lung studies seem to be accurate in terms of lesion detection and screening, risk stratification, nodule segmentation, radiogenomic analysis, prognosis prediction, treatment planning (volume, morphology, relationships of the lesion) and response, and starting to play a pivotal role in oncological imaging (Figure 2).

The availability of open-source image datasets has enabled the development and validation of effective CAD tools (Figure 3).

Chi et al. developed a chest CT pulmonary nodule detecting system based on a deep convolutional neural network (CNN) framework. LUNA16 and Ali Tianchi were the databases used for training and testing this system. The peculiarity of this approach was that the framework consisted of three cascaded networks: a U-Net for segmentation of the region of lung parenchyma from chest CT, a modified U-Net network, to identify suspicious nodules, and a modified U-Net to detect the true pulmonary nodule. A precision of 0.8792, a sensitivity of 0.8878, and a specificity of 0.9590 were obtained [103]. The highlighted limitation was that, when the nodule was low-density and placed in the lung parenchyma edges, the method had problems in nodule distinction from the outside-lung region.

Also, Khosravan et al. used a CNN framework, called S4ND, for three-dimensional imaging which permits to identify in a faster way lung nodules. LUNA16 dataset was used and the system showed a sensitivity of 95.2% [104]. The strength was that S4ND did not need any further post-processing or user guidance to refine detection results

Nasrullah et al. developed a detecting and classification system for lung nodules in low-dose chest CT based on two deep three-dimensional customized mixed link networks (CMixNet). LIDC-IDRI dataset was used. The system showed a sensitivity of 94% and specificity of 91% [105]. The strength of this study was that the nodules were not only detected but also further analyzed for classification as benign or malignant. Moreover, the AI-based nodule classification was further evaluated with patients-related factors, such as symptoms, age, smoking history, clinical biomarkers, and nodules to reduce false positives and misdiagnosis.

Other authors used the LUNA16 dataset to develop only one deep learning model, but also combined the performance of two-dimensional CNNs, and also applied data augmentation to rotate the images in all possible directions and create more copies of the same data from different angles. The proposed method provides an accuracy of 95% [106].

Cai et al. also adopted the MaskRCNN model together with a feature pyramid network to extract feature maps. The system was used to identify suspected lung nodules. LUNA16 dataset was used and a sensitivity of 88.70% was obtained. In this study, the authors provided not only detection but also segmentation and 3D visualization systems [107].

Manickavasagam et al. used LIDC/IDRI images to develop a convolutional neural network with 5 convolutional layers. The system was based on image feature extraction and characterization and reached accuracy, sensitivity, specificity, and area under the roc curve of 98.88%, 99.62%, 93.73%, and 0.928, respectively [108]. Despite these results, the CAD system still has several limitations. The main one is the high number of false positive results related to blood vessels or other soft tissue structures that are misinterpreted by the system itself. This aspect reduces the accuracy and efficacy of CAD screening tools in large populations [109].

Effective nodules classification techniques can reduce the false positive rates: Tran et al. developed a system dividing nodules and non-nodules with an accuracy of 97.2%, sensitivity of 96.0%, and specificity of 97.3%. Also, other authors tried to reduce false positive results such as Wu et al. [110] which reached an average accuracy of 98.23% and a false positive rate of 1.65%, and Mastouri et al. [111] which achieved an accuracy rate of 91.99%.

After lung nodule detection, the nodules should be characterized as benign or malignant. CNNs are used to analyze various lung nodule features such as morphology, shape, and growth rate, for this purpose [95]. Zhang et al. [112], Al-Shabi et al. [113], and Liu et al. [114] proposed three different systems for this kind of analysis achieving classification accuracy, respectively, of 92.4%, 92.57%, and 90%.

In future perspectives, radiomics and radiogenomics analysis will be available allowing personalizing patient profile, therapy, and prognosis but nowadays no clear results in terms of accuracy are available.

### 4.4. Nodule Segmentation

A precise lung nodule segmentation is difficult due to size, which often is small, and position, in particular when nodules are near the lung’s edges or vessels.

Segmentation systems may vary in terms of architecture, image pre-processing techniques, and training strategies [95]. Different approaches are currently available for nodule segmentation with multiview or general neural network architecture.

Multiview approaches consider multiple perspectives of lung nodules and combine them as input for neural networks.

On the other hand, general neural network architecture builds upon traditional CNNs by modifying or adding certain components.

Lung nodule type and shape influence the choice of segmentation method and different studies have combined conventional convolutional and additional neural network CNN architecture to reach better lung nodule segmentation.

Two commonly used basic structures are the U-Net and Fully Convolutional Neural Networks (FCN) architectures.

Numerous works have demonstrated the effectiveness of convolutional neural networks, especially FCN and U-Net, in enhancing lung segmentation performance. These networks follow a two-step process.

First, they extract image feature maps by down-sampling to filter out irrelevant information while preserving important details. Second, they upsample the resulting feature maps to achieve a higher-resolution display.

Different researchers have found inspiration from these networks and decided to modify their models.

For example, Huang et al. used a customised FCN for analyzing different lung nodule aspects such as detection, merging, false positives, and segmentation.

Their model achieved an average Dice Similarity Coefficient (DSC) of 0.793 on the LIDC-IDRI dataset [115].

Usman et al. introduced a dynamic modified ROI (Region Of Interest) algorithm, using the Deep Res-UNet architecture to improve segmentation and better-localizing lung nodules and better calculate their volumes. This was a two stages approach: at first training and predicting the axial axis using Deep Res-Net, focusing on the new ROI, and training the network with Deep Res-UNet architecture as the second stage. An average DSC of 87.55%, 91.62% sensitivity (SEN), and a positive predictive value (PPV) of 88.24% were achieved [116]. The main limitations of the study were related to the strict connection of nodules to non-nodule structure which made the definition of the nodule margins difficult and to the small diameter of the nodules. Zhao et al. implemented a patch-based 3D U-Net and contextual CNN for automatic segmentation and classification of lung nodules. In this study, the 3D U-Net model was applied and implemented with generative adversarial networks (GANs) and contextual CNN to reduce false positives and improve nodule classification. This method demonstrated good results in both segmentation and classification tasks [117]. The authors developed a multi-step process including nodule segmentation first, followed by classification into benign or malignant, all integrated into a complete automatic algorithm for lung nodule detection. Kumar et al. utilized the V-Net architecture for lung nodule segmentation, focusing on the convolutional layers and omitting the pooling layers. The authors aimed to test the potential of a novel 3-dimensional CNN that was developed to segment prostate MRI volumes, instead of a 2-dimensional CNN. Their model was able to process different types of scans and achieved a high DSC of 0.9615 on the LUNA16 dataset [118]. This model can allow better visualization and classification of lung nodules.

Keetha et al. integrated U-Net with Bi-FPN developing a resource-efficient U-Det architecture. LUNA dataset was used to train and test their network which achieved an average DSC of 82.82%, SEN of 92.25%, and PPV of 78.92% [119]. The strength of the U-Det model was the ability to segment challenging cases such as cavitary nodules, ground glass nodules, and small and peripheral nodules.

Numerous research studies have introduced novel architectures that leverage multiple views of lung nodules as inputs to neural networks, leading to improved segmentation results.

These methods primarily rely on convolutional neural network (CNN) structures and incorporate multiscale or multiview techniques for network training.

For example, in the study of Zhang et al., a conventional approach for nodule segmentation by employing a multiscale Laplacian of Gaussian filter was adopted. Their method achieved a detection score of 0.947 on the LUNA16 dataset [120].

The study from Cao et al. presents a dual-branch residual network (DB-ResNet), integrating multiview and multiscale CT nodules features with CNN intensity features, obtaining a better performance in lung segmentation, with an average sensitivity (SEN) of 89.35%, and an average DSC of 82.74% [121]. The proposed model can simultaneously capture multiview and multiscale characteristics of lung nodules and showed an average DSC higher than that of human experts.

Wu et al. developed a multitask U-Net architecture-based model, called PN-SAMP (Pulmonary Nodule Segmentation Attributes and Malignancy Prediction) with an average DSC of 73.89% and an average SEN of 97.58%. The authors aimed not only to provide a segmentation system but also to estimate the risk of nodule malignancy. [122].

Lastly, Wang et al. proposed a two-stage central-focused CNN (CF-CNN) lung nodule segmentation model, tested on the LIDC-IDRI dataset. This model showed an average DSC of 82.15% and an average SEN of 92.75% [123]. The advantage of this model was that it captured nodule-sensitive characteristics from 3-D and 2-D images simultaneously.

As previously mentioned, pulmonary nodules exhibit various types, shapes, and clinical characteristics; therefore, the detection procedures and associated challenges vary from case to case. In general, detecting nodules located near blood vessels and the pleura poses the greatest challenge, therefore, all models prioritize enhancing the boundaries of the nodules.

Pezzano et al. proposed a U-Net-based network for lung nodule segmentation, using the public LIDC-IDRI dataset, and introducing the multiple convolutional layers (MCL) module. This model permitted us to better define boundaries, morphological aspects, and edges of the nodules [124]. The advantage of this solution was the reduction in the image resolution loss with increased detail on the nodule margins.

Voxel and shape heterogeneity (VH) and SH properties were included in a model created by Dong et al. While SH concentrates on the features of nodule shape, VH captures variations in grey voxel values. The researchers discovered that whereas SH was excellent at catching border information, VH learned grey information effectively [125]. The model is based on a lung volume cube. The current limitation is the need to shift from cubes containing lung nodules to a whole 3D volume. A possible solution is that the radiologist decides the position of the cube to assist the model in 3D segmentation.

Cao et al. introduced the DBResNet model, which preserved intensity features only of the target voxel, this layer was called the central intensity-pooling layer (CIP) and is optimal in juxta-pleural and small nodules evaluation, [121].

The performance of the aforementioned techniques was only somewhat worse for nodules with juxta-pleural, juxta-vascular, and ground glass opacity. AI-Shabi et al. introduced non-local blocks to capture global features and residual blocks with a 3 × 3 kernel size to extract local features to overcome this limitation. This approach minimized the number of parameters while achieving high performance. The model was trained and tested on the LIDC-IDRI dataset, yielding outstanding results compared to DenseNet and ResNet, with an area under the curve (AUC) of 95.62% for transfer learning [126].

### 4.5. Nodule Characterization

Pulmonary nodules are low-contrast tissues that are not easily distinguished from their surroundings. Strategies such as chest radiography [127] and sputum cytology demonstrated limited results in screening programs; low-dose chest CT, instead, can detect potentially malignant lung nodules of a few millimetres in size in high-risk populations, reducing lung cancer mortality [19,128,129].

The most prevalent CT appearance of early-stage lung cancer is opacities in the lung parenchyma that are not considered normal anatomy, more often known as pulmonary nodules. The discovery of all pulmonary nodules is the initial step in the workflow toward lung cancer diagnosis [130]. Once a nodule has been identified and defined as a nodule, it should be classified as benign or malignant.

AI can analyse and interpret complex medical data, thus, aiding in the diagnosis, management, and prediction of treatment outcomes in different clinical presentations [129,131].

Radiomic introduces a quantitative evaluation in the world of qualitative radiological image interpretation and assessment [132]. Radiomics' early uses explored its potential in rheumatic heart and pulmonary diseases on chest radiographs, but nowadays it is becoming important in the oncologic field [133].

Radiomics has developed as an offshoot of the wider “omics” fields in molecular biology, but unlike the other “omics” (genomics, proteomics, and transcriptomics), is based on radiological imaging analysis and extraction of a huge number of quantitative features, rather than invasive biopsy or molecular assays [134].

Quantitative image features are considered non-invasive biomarkers that reflect the underlying tumour pathophysiology and heterogeneity [135,136].

Based on computer algorithms, it processes many imaging modalities (including ultrasound, CT, PET, MRI, and traditional radiology) by analyzing the chosen region of interest or segmented volume [137,138] to obtain multiple features using data-characterization algorithms. Such features assist in identifying cancer characteristics hidden from the naked eye of a human expert [139,140].

Additionally, these imaging features can effectively produce a distinct phenotypic atlas for each tumour [75]. The development of novel, repeatable, image-based biomarkers that have been prognostic for clinical outcomes, including overall survival and distant metastases, has been made possible by the association of clinical data with this atlas [141,142].

Most radiomics studies on lung cancer are retrospective and based on conventional medical images obtained from routine clinical protocols [143] with the accuracy of nodule delineation and segmentation affected by the CT image quality. Consequently, the performance of our technical approach to quantitative radiomics analysis and machine learning is dependent on the image quality of thin-section CT [144].

The bulk of medical imaging segmentation algorithms now available rely on “region growing” techniques thorough the connection of voxels over a specified threshold [145]. Even if solid intraparenchymal pulmonary nodules are easy to identify, this task becomes difficult when vessels, airways, or pleura are near lung nodules due to their similar attenuation. In these cases, structure removal through image processing analysis with morphological criteria becomes useful. Subsolid nodules are more difficult to segment.

Image segmentation for feature extraction can be manual, semi-automatic (through specific algorithms), or completely automatic (thorough deep learning algorithms) [146]. Manual and semi-automatic (with manual correction) are the most frequently used approaches, despite these needing time and being influenced by observer bias. For this, it is important to check the intra- and inter-observer reproducibility of the obtained radiomic features and remove non-reproducible features from the analysis. Instead, deep learning-based image segmentation is still in the developing phase and needs to be improved, despite several different algorithms being available [147,148].

The first major use of AI methods in the characterization of pulmonary nodules is thus lung nodule segmentation.

Accurate segmentation of lung nodules is critical as it allows for automatic volume estimation. These measurements are used to decide how to handle the nodule in accordance with international recommendations [149,150,151,152,153], as well as to calculate the volume doubling time when a second follow-up CT examination is conducted. Some AI techniques for lung cancer screening can not only detect and quantify lung nodules but also recommend Lung-RADS classification. This is accomplished by combining the mean nodule diameter measurement and the nodule type. CNNs, which are deep neural networks based on a series of inputs, are the most commonly used deep learning algorithms for image processing [154,155].

To obtain various types of features, such as first-order histogram features, second-order texture features, and higher-order texture features, there are numerous methods and formulas available. Univariate and multivariate statistical models can be used to analyze the feature selection process [156]. Ma et al.’s study [157] demonstrated the potentially huge number of radiomic textural features that can be retrieved from lung nodules.

Tu et al. analyzed thin-section CT images and extracted quantitative features from them, discovering that radiomic analysis of grey-scale intensity is useful for differentiating benign and malignant nodules [144].

Chae et al. used a combination of texture-based features to differentiate pre-invasive from invasive lung adenocarcinoma (area under the curve, 0.981) in a study including 86 part-solid ground glass nodules [158].

Pérez-Morales et al. estimated radiomic properties from the intratumoral and peritumoral regions, using these to develop a model that estimated lung cancer patients’ outcomes when the tumour was identified during screening [135].

Model development is the final step. Building a radiomics model entails three major steps: selecting radiomic characteristics, selecting the training cohort and machine learning models, and finally validating the test cohort. Model performance is often quantified in terms of calibration and discrimination, and it is evaluated using the c-index or the AUC [156]. The trained model is tested on new independent data to see how well it performs. If the model performs well on the validation data and performs well on the training data, its robustness and generalization are validated. Assuming the training data set is representative, lower prediction performance on validation data would imply overfitting (where a model draws incorrect conclusions on training data that are not applicable to new observations). Underfitting (where the classification model is unable to derive meaningful conclusions from the data) would be indicated by poor performance on both the training and validation data [156].

Huang et al. introduced a radiomic nomogram including radiomic signatures from nodular and peri-nodular areas in order to distinguish preoperatively PILs (invasion lesions) from early-stage pulmonary interstitium ILs (pre-invasive lesions). In particular, peri-nodular radiomic signature combined with nodular radiomic signature increases the ability to distinguish between pulmonary interstitium ILs and PILs [159].

On data from patients who had undergone surgery for stage I NSCLC, Yu et al. created a radiomics signature to predictive the mortality risk after first-line treatment [160].

The literature shows that radiomics is being used for the categorization of the analysed nodule into one of two groups: malignant or benign, but it has been applied in several other classifications and characterization of lung nodules.

One example can be the prediction of the response to therapy (primarily radiotherapy) Coroller et al. found that a radiomic signature could be used to predict the response to chemoradiotherapy in NSCLC patients [141].

In order to identify individuals who are more likely to benefit from PD-1/PD-L1 inhibitors in the setting of advanced or recurrent NSCLC, Cousin et al. conducted a study to identify CT-based delta-radiomics signature [161].

Accurately classifying tumour aggressiveness is important in assessing prognosis, treatment, and follow-up [162,163].

Hou et al. constructed a combined deep learning model with clinical and radiomic features for NSCLC patients’ survival prediction. [164].

Quantitative imaging can help determine the presence or spread of cancer in different organs or tissues [165].

Dou et al. demonstrated that peritumoral rim radiomic features are significantly associated with distant metastasis from lung adenocarcinoma [166].

#### Virtual Biopsy

In solid tumours, genes, proteins, cells, microenvironment, tissues, and organs all exhibit spatial and temporal heterogeneity [167,168]. This restricts the use of biopsy-based molecular assays but, in contrast, creates a significant opportunity for non-invasive imaging, which can be used to non-invasively capture intra-tumoral heterogeneity [169,170].

Virtual biopsies are currently being investigated mostly in the scientific setting [166]. Oncology is seeing a surge in new applications. The introduction of targeted immunotherapies for lung cancer has radically transformed the therapy landscape and survival rates [171]. As a result, identifying targetable mutations and expression levels has become an important step in patient management and provides another justification, in addition to confirming the histologic cancer subtype, for doing a physical tissue biopsy [172].

Imaging biomarkers provide unique insight into tumour behaviour and therapy response [173]. These are now playing an important part in the medication development process. Radiogenomics and radioproteomics have the aim of associating imaging phenotypes to genetics and protein expression patterns of a person through bioinformatics approaches [174].

Radiogenomics concerns the relation between radiology and genomics, non-invasively researching biological features related to clinical outcomes, with the purpose of replacing surgical biopsies and histopathologic analysis, helping the evolution towards a personalized medicine system [166,175].

At the Mayo Clinic in Rochester, Minnesota, Lee et al. created a machine learning method called Computer-Aided Nodule Analysis and Risk Yield (CANARY). CANARY discovered nine distinct exemplars (radiomic fingerprints) that define the lung cancer spectrum. CANARY, as a virtual biopsy technique, has been demonstrated to correlate directly with adenocarcinoma invasion [176].

The radiomics signals derived by Nair et al. from diagnostic CT and PET-CT images have the potential to be used as biomarkers to predict EGFR mutations in NSCLC [177].

Preliminary results from Lafata et al. show that an integrated radiomic, cfDNA, and ctDNA liquid biopsy analysis in patients with locally advanced lung cancer is feasible. Their research found that tumours that appeared more homogenous and attenuated on CT imaging had detectable ctDNA TP53 mutations and static changes in cfDNA content early in therapy [178].

The necessity for radiophenotyping for accurate patient classification is stronger than ever in the era of precision medicine [179]. Radiomics characteristics and signatures, ideally, can be used as imaging biomarkers. The quality and sophistication of published radiomics studies are increasing, resulting in a plethora of new radiomics-based findings in the field of lung cancer. The next challenge for radiologists will be to stay up with the rapid advancement of technology.

## 5. Future Perspectives

Nowadays AI algorithms are still difficult to create and validate due to the disorganized curation of imaging and clinical data. A forward step in these terms is still a challenge, but necessary for a better future use of AI.

Thoracic imaging and thoracic oncology particularly benefit from the development of AI, classical machine learning, and deep learning methods. In recent years, AI-based lung cancer detection [180], characterization, and risk stratification have demonstrated excellent performance, even comparable to that of histopathology [144].

Automating time-consuming, repetitive tasks like finding image-based biomarkers and checking for lung nodules will eventually be possible thanks to AI. With these advancements, personalized medicine will hopefully be possible, enabling non-invasive, repeatable disease characterization that will improve therapeutic management.

Platforms to choose between different AI applications, and the integration of AI technology into image archiving and communication systems, are needed to bring AI into routine practice [181].

The future direction is to integrate AI applications for lung cancer screening in a unique path to optimize the screening procedure. The first step will be represented by pre-screening applications for a personalized risk assessment to optimize patients’ eligibility criteria. The second step consists of image acquisition with low-dose protocols, reconstructed through deep learning-based algorithms, in order to maintain an optimal image quality with limited radiation exposure.

The third step is represented by an AI-based system for automated nodule detection to reduce the radiologist's workload. This process is followed by the phase of nodule characterization as benign or malignant to resource utilization, costs, and chance of unnecessary biopsy or surgery.

## 6. Conclusions

AI has the potential to truly revolutionize the early detection of lung cancer. Its possible fields of applications are wide, including image reconstruction, personalized screening programs, automatic nodules detection, segmentation, and characterization. The integration of multimodality and the creation of tested and validated efficient models may permit an accurate characterization and evaluation of lung nodules and of patients’ outcomes or survival. To reach this, collaboration, cooperation, and integration between radiologists and clinicians is necessary. AI has huge potential clinical applications and the AI usage in chest imaging is one of the more motivating challenges of future medicine.

## Figures and Tables

**Figure 1 cancers-15-04344-f001:**
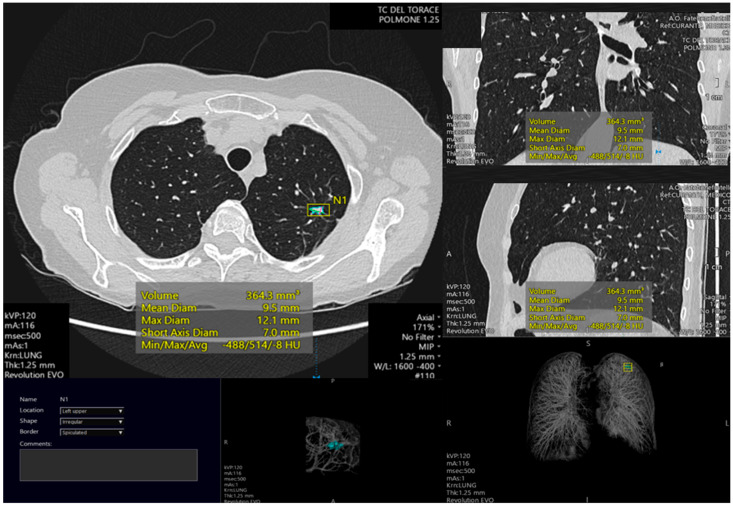
Example of pulmonary lesion automated segmentation. The lesion is located in the left higher lobe. The automated analysis permits to calculate different lesion parameters such as volume (mm^3^), mean diameter (mm), maximum diameter (mm), short axis diameter (mm) and density (Hounsfield Units). Also, 3D reconstruction is shown.

**Figure 2 cancers-15-04344-f002:**
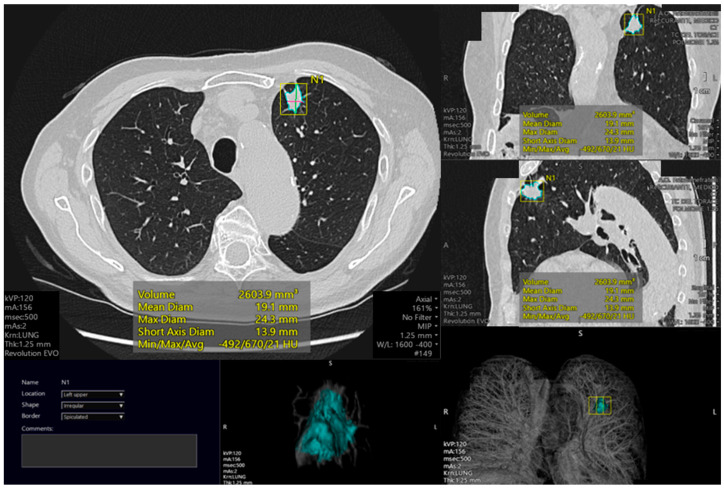
Example of left upper lobe pulmonary lesion with automated analysis. The automated study permits to calculate different lesion parameters such as volume (mm^3^), mean diameter (mm), maximum diameter (mm), short axis diameter (mm), and density (Hounsfield Units). A circle mark surrounding the lung nodule is realized in the identified lesion in order to analyze it. Also, 3D reconstruction is shown.

**Figure 3 cancers-15-04344-f003:**
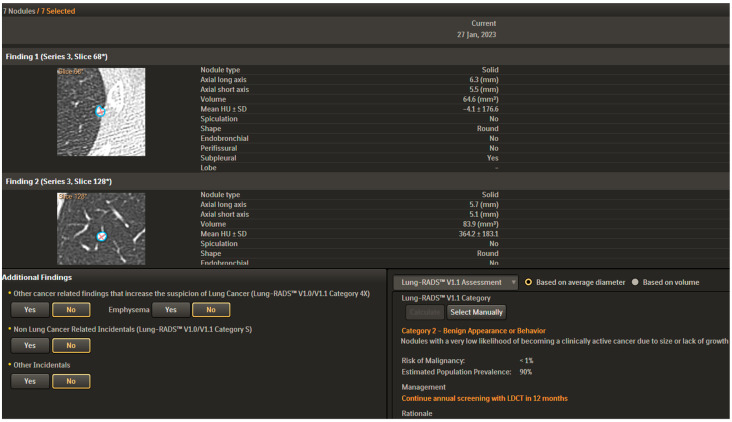
CAD analysis is present in the lower part of the figure. Nodule type, dimensions, volume, mean HU, speculation, shape, side, and site are described. Correlation with LUNG-RADS classification is realized.

## Data Availability

The data presented in this study are available in this article.

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
