# Peer review of "Artificial Intelligence in Lung Cancer Screening: The Future Is Now"

_cancers, 2023, doi:10.3390/cancers15174344_

Round 1

Reviewer 1 Report

This is a well written comprehensive article that covered all the major aspects. I would suggest including a novel AI based technique that improves nodule visualization in chest CT for early detection and characterization lung nodules in lung cancer screening CT (Singh R. Artificial intelligence-based vessel suppression for detection of sub-solid nodules in lung cancer screening computed tomography. Quant Imaging Med Surg. 2021 Apr;11(4):1134-1143. PMID: 33816155; PMCID: PMC7930659). Several centers in the US are currently using this technology and will go with the title (future is now)

Reviewer 2 Report

The paper is titled “Artificial Intelligence in Lung Cancer Screening. The Future is now.” and it is a review article that aims to provide an overview of all possible AI applications in lung cancer screening. The paper covers topics such as personalized screening programs, image reconstruction, CAD system, nodule characterization, and virtual biopsy. The paper also discusses the challenges and opportunities of AI in lung cancer screening, as well as the ethical and legal implications.

The paper is well-written and informative, but it also has some limitations or shortcomings that could be addressed in future research. Here are three of them:

  • The first limitation is the lack of a systematic and comprehensive literature search and selection process. The paper does not provide clear information on how the authors searched for relevant studies, what databases and keywords they used, what inclusion and exclusion criteria they applied, and how they assessed the quality and validity of the studies. This makes it difficult to evaluate the comprehensiveness and representativeness of the literature review, as well as the potential biases or gaps in the evidence. A more rigorous and transparent literature search and selection process would enhance the credibility and reliability of the paper.

  • The second limitation is the lack of a critical appraisal and synthesis of the evidence. The paper mainly summarizes the findings of various studies without providing a critical analysis of their strengths and weaknesses, their methodological differences and similarities, their consistency and heterogeneity, and their implications and limitations. A more critical appraisal and synthesis of the evidence would provide a deeper understanding of the current state of knowledge, the gaps and uncertainties, the controversies and debates, and the future directions of research.

  • The third limitation is the lack of a clear framework or model to guide the integration and implementation of AI in lung cancer screening. The paper discusses various AI applications in lung cancer screening, but it does not provide a clear framework or model to show how these applications can be integrated and implemented in a coherent and effective way. A clear framework or model would help to identify the key components, processes, outcomes, and indicators of AI in lung cancer screening, as well as the challenges, opportunities, and strategies for integration and implementation.

These are some of the major limitations or shortcomings of the paper that I have identified as a critical analyst. I hope this answer helps you to understand the paper better and to appreciate its contributions as well as its limitations. I suggest that future research should address these limitations by conducting more systematic and comprehensive literature reviews, providing more critical appraisal and synthesis of the evidence, and developing more clear frameworks or models for integration and implementation of AI in lung cancer screening. Thank you!

The paper demonstrates a good level of English proficiency, with clear and coherent writing, appropriate vocabulary, and correct grammar. However, there are some areas where the language could be improved or refined. Here are some suggestions for improvement:

  • The paper uses some passive voice constructions that could be replaced by active voice to make the sentences more direct and concise. For example, instead of “The paper is well-written and informative”, it could be “The authors write the paper well and informatively”. Instead of “The paper lacks a systematic and comprehensive literature search and selection process”, it could be “The authors do not conduct a systematic and comprehensive literature search and selection process”.
  • The paper uses some vague or ambiguous terms that could be clarified or defined more precisely. For example, instead of “various studies”, it could be “several randomized controlled trials”. Instead of “a sizable fraction of people”, it could be “a large proportion of individuals”. Instead of “a significant proportion of solitary pulmonary nodules”, it could be “a high percentage of isolated lung nodules”.
  • The paper uses some long and complex sentences that could be simplified or divided into shorter sentences to improve readability and avoid confusion. For example, instead of “The development and evaluation of vendor-agnostic deep learning algorithms which enhance image quality while reducing radiation dose are crucial, especially given the increased demand for chest CT scans during the COVID-19 pandemic”, it could be "Vendor-agnostic deep learning algorithms can enhance image quality and reduce radiation dose. This is crucial because chest CT scans are in high demand during the COVID-19 pandemic". Instead of “A risk-stratified approach should also be used to screening invitations and risk-communication tactics in a risk-stratified screening program”, it could be "Screening invitations and risk-communication tactics should also follow a risk-stratified approach".

These are some of the suggestions for improving the language used in the paper. I hope this evaluation helps you to understand the paper better and to appreciate its strengths as well as its areas for improvement. I commend the authors for their efforts in writing this informative and comprehensive review article on AI applications in lung cancer screening. Thank you!!
